# Gliricidia Agroforestry Technology Adoption Potential in Selected Dryland Areas of Dodoma Region, Tanzania

**Martha Swamila** [1,*], **Damas Philip** [1], **Adam Meshack Akyoo** [1], **Stefan Sieber** [2,3], **Mateete Bekunda** [4] **and Anthony Anderson Kimaro** [5] 

1   School of Agricultural Economics and Business Studies, The Sokoine University of Agriculture, Morogoro P.O. Box 3007, Tanzania; philip@sua.ac.tz (D.P.); akyoo63@sua.ac.tz (A.M.A.)

2   The Leibniz Centre for Agricultural Landscape Research (ZALF), Eberswalder Str. 84, 15374 Müncheberg, Germany; stefan.sieber@zalf.de

3   Department of Agricultural Economics, Faculty of Life Sciences Thaer-Institute, Humboldt-Universität zu Berlin, Unter den Linden 6, 10099 Berlin, Germany

4   International Institute of Tropical Agriculture (IITA), Duluti, Arusha P.O. Box 10, Tanzania; m.bekunda@cgiar.org

5   ICRAF-Tanzania Country Programme, World Agroforestry (ICRAF), Dar es Salaam P.O. Box 6226, Tanzania; a.kimaro@cgiar.org

*   Correspondence: marthaswamila@yahoo.com

**Abstract:** Declining soil fertility is one of the major problems facing producers of field crops in most dryland areas of Sub-Saharan Africa. In response to the declining soil fertility, extensive participatory research has been undertaken by the World Agroforestry (ICRAF) and smallholder farmers in Dodoma region, Tanzania. The research has, amongst others, led to the development of Gliricidia agroforestry technology. The positive impact of Gliricidia intercropping on crop yields has been established. However, information on farmers' willingness and ability to adopt the Gliricidia agroforestry technology on their farms is limited. This study predicts the adoption of Gliricidia agroforestry and conventional mineral fertilizer use technology. Focus Group Discussions (FGDs) were conducted with groups of farmers, purposively selected based on five sets of criteria: (i) at least 2 years of experience in either trying or using Gliricidia agroforestry technology, (ii) at least 1 year of experience in either trying or using the mineral fertilizer technology (iii) at least 10 years of living in the study villages, (iv) the age of 18 years and above, and (v) sex. The Adoption and Diffusion Outcome Prediction Tool (ADOPT) was used to predict the peak adoption levels and the respective time in years. A sensitivity analysis was conducted to assess the effect of change in adoption variables on predicted peak adoption levels and time to peak adoption. The results revealed variations in peak adoption levels with Gliricidia agroforestry technology exhibiting the highest peak of 67.6% in 12 years, and that the most influential variable to the peak adoption is the upfront cost of investing in Gliricidia agroforestry and fertilizer technologies. However, in Gliricidia agroforestry technology most production costs are incurred in the first year of project establishment but impact the long term biophysical and economic benefits. Moreover, farmers practicing agroforestry technology accrue environmental benefits, such as soil erosion control. Based on the results, it is plausible to argue that Gliricidia agroforestry technology has a high adoption potential and its adoption is influenced by investment costs. We recommend two actions to attract smallholder farmers investing in agroforestry technologies. First, enhancing farmers' access to inputs at affordable prices. Second, raising farmers' awareness of the long-term environmental benefits of Gliricidia agroforestry technology.

**Keywords:** ADOPT; adoption; Gliricidia agroforestry; soil fertility; dryland areas

## 1. Introduction

Sustainable crop productivity in many dryland areas of Sub-Saharan Africa (SSA) is limited by sharply declining soil fertility [1–4]. In Tanzania, the impact of declining soil fertility on crop productivity is critical in the dryland areas, such as parts of the Dodoma region, where low soil fertility has been often cited as one of the major on-farm production constraints [5]. The productivity of most crops in the region is low, for example, maize yields, are reported to range from 1 to 1.5 tons/ha. This is below farmer average estimated maize productivity potential of 4–4.5 tons/ha [6]. Consequently, low crop productivity contributes to low farm income and a high level of food poverty line at 36% and up to 51% based on expenditures [6]. Declining agricultural productivity in SSA is further exacerbated by negative impacts of climate change, including shifting seasons and extended periods of drought [7,8].

The degradation of soil fertility is caused by two major inter-related factors. The first factor is the breakdown of natural fallow soil fertility restoration methods due to an increase in the human population and consequently reduced per capita land availability. The increasing population has also contributed to the degradation of natural resources as the agricultural production continues to encroach into forests and woodlands in response to declining productivity on farmlands. The second factor is non- or sub-optimal use of mineral fertilizers by a large majority of smallholder farmers due to high prices and limited availability. The situation became more critical following the failure of the implementation of subsidy programs in many countries such as Tanzania and Zambia [9,10].

As a way to deal with soil fertility and climatic challenges and increase agricultural productivity, researchers from the World Agroforestry (ICRAF) and small scale farmers involved in the production of cereal food crops validated Gliricidia agroforestry technology in the dryland areas of Kongwa and Chamwino districts in Dodoma region. Researchers' Designed Farmers' Managed (RDFM) trials including Gliricidia agroforestry technology began in 2015 at research stations. Additionally, smallholder farmers involved in managing trials at the research stations tested the technologies on their farms under their design and management. Farmers were involved in the management of research trials to demonstrate the real field situation and enhance their knowledge on the tested technologies for subsequent adoption of the recommended practices [11,12].

The biophysical assessment results show the increased yields in Gliricidia intercropped trials. For instance, it was established that, intercropping maize with Gliricidia and pigeon pea improved grain yields by up to 33%, besides fodder and wood supply for improved livestock nutrition and household energy [13]. However, the major question is, what proportion of farmers will be willing and able to adopt Gliricidia agroforestry technology on their farms? A study conducted in Cameroon, Zambia, and Kenya noted variations in the adoption potential of agroforestry technologies, ranging from moderate to high [14]. However, it only employed qualitative approaches and did not specify the rates corresponding to the moderate and high adoption potential ranges. Moreover, adoption of improved agricultural technologies such as agroforestry is context and site-specific, as might be temporarily and spatially affected by factors such as climate, soils, and availability of resources including land and labor [5,15–17]. So far, the adoption prediction studies conducted in Tanzania and elsewhere have focused on introduced chicken strains [18]; fertilizer micro-dosing and rain-water harvesting technologies [5]; improved fodder crops [19] and improved crop varieties [20]. According to [21] and [20], the adoption of technologies in the communities, among others, is influenced by the specific characteristics of the technology such as the relative advantage and learnability. Efforts to predict the adoption of agroforestry technologies including Gliricidia intercropping are scanty. Therefore, it is plausible to extend assessment on predicting farmers' likelihood to adopt the newly developed Gliricidia agroforestry technology focusing on the agro-ecologies of the dryland areas. The prediction of Gliricidia agroforestry technology adoption would increase the understanding of factors influencing its adoption in the study communities [22–24].

The present study employed the Adoption and Diffusion Outcome Prediction Tool (ADOPT) to predict the adoption of (i) Gliricidia agroforestry technology (ii) mineral fertilizer use, representing the conventional soil fertility management technologies, and (iii) assess the effect of changes of the ADOPT

population and technology factors on peak adoption level and time to peak adoption. The findings of the study were aimed to inform policy and extension efforts in enhancing farmers' adoption of Gliricidia agroforestry and mineral fertilizer technologies, in appropriate combination, for increased farm productivity and profitability.

## 2. Methodology

### 2.1. Theoretical Framework

Several diffusion of innovations' theories have been used to study adoption of improved agricultural technologies. Rogers [21] and Bass [25] provide useful approaches to assess adoption of technologies, and describe the adoption process focusing on five groups of adopters: innovators (2.5% of people who are ready to take risks by trying out innovations), early adopters (13.5% of people, referred as opinion leaders who try new things with caution), early majority (34% of people who are careful but quick to change), late majority (34% of people who adopt new ideas after the majority), and laggards (16% of people who are traditional, conservative, and slow to change). According to [21], it takes time for innovation to diffuse through society. Thus, efforts to enhance adoption should start by convincing innovators and early adopters expecting other groups of adopters to join in the future. Rogers [21] theory was assumed to apply to the study on adoption of Gliricidia agroforestry technology in dryland areas of Kongwa and Chamwino districts in Dodoma region. Factors influencing the diffusion and adoption of agricultural improved technologies such as the relative advantage and trialability are incorporated in ADOPT to predict the peak adoption level and the corresponding time in years.

Bass [25], contends that adoption of innovations is the result of interactions between potential and actual adopters. Bass [25] theory applies to this study, as the group involvement which determines the degree of interactions between potential and actual adopters of Gliricidia agroforestry technology, is integrated in adoption predictions. Additionally, previous studies on adoption predictions [18,26] used Bass [25] theory to estimate probabilities of adopting new agricultural technologies among smallholder farmers in Tanzania and elsewhere.

### 2.2. Nature of Technology Change

The adoption of improved agricultural technologies such as agroforestry is associated with two major properties. The first property is the shift of production function as the output is increased with adoption of the improved technologies, here referred as input (Figure 1) from OY1 to OY2 other factors remaining the same [27]. Concerning adoption of Gliricidia agroforestry technology, a 33% increase in grain yields was attained in the selected study sites as the result of improved soil fertility [13].

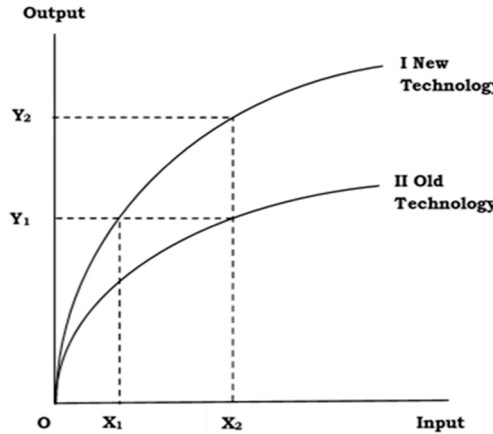

**Figure 1.** Adoption of technology and impact on productivity. Source: [27].

The second property is the increase in monetary discounted profits or the decrease in production costs. A rational decision making producer will only adopt the technology that will maximize farm productivity and profitability with the lowest possible cost combination of inputs [28].

Improved agricultural technologies such as agroforestry that increase farm productivity and profitability can have three features at the farm level. These features are factor-saving [6], factor-using [10], and output-increasing [13]. Gliricidia agroforestry technology is expected to save up to 50% of mineral fertilizer purchasing costs (factor saving). The use of Grilicidia agroforestry technology utilizes more labor in various managerial aspects such as preparation of seedbed, watering of trees, transplanting, preparation of plot layouts, and pruning (factor using). However, the application of this technology is expected to increase crop productivity as applied to the most improved agricultural practices, and hence output-increasing [29].

### 2.3. Empirical Review

The adoption of agricultural technologies in the context of smallholder farmers has been well studied in Tanzania and elsewhere. Several studies have been conducted including those on the adoption of improved crop varieties [30–33]; fertilizer technologies [34,35]; conservation agriculture technologies [36]; irrigation schemes [37] and agroforestry systems [14,38,39]. However, most of these studies are qualitative and limited to ex-post evaluations, employing regression analyses. Among others, a major shortcoming encountered by the ex-post evaluation approach, is the low contribution of knowledge in the project designing and implementation phases.

Approaches used to predict adoption of technologies include surveys of producer intention [40–42], expected profit [43–45], historical trend [46], and ADOPT [5,18–20]. The expected profit approach uses farm-level financial data to predict adoption based on profitability. However, it ignores non-profit factors such as risks and environmental costs and benefits influencing the adoption of new practices. The historical trend approach predicts adoption through extrapolation. However, it requires the presence of similar technology under consideration.

The ADOPT makes quantitative predictions and complements information from other qualitative approaches and farm decision tools, including farm surveys as well as economic and biological simulation models. [20] used ADOPT to predict adoption of the following practices in Australia: (a) autosteer (GPS guidance in tractors), (b) growing insect-resistant (Bt) transgenic cotton, (c) growing a new species of legume crop, (d) growing a new wheat variety, Mace, (e) using no-till cropping, and (f) planting saltbush forage shrubs. The results showed variations in peak adoption levels ranging from 9% (saltbush) to 98% (autosteer) in 9–23 years, respectively. [5], employed ADOPT to simulate the adoption of fertilizer micro-dosing and rainwater harvesting technologies using tied ridges. They noted the high peak adoption levels of between 90% and 94% to be attained in 10.5–11.5 years for fertilizer micro-dosing without and with tied ridges, respectively. [18] used ADOPT to simulate the peak adoption levels of introduced chicken strains in Bahi, Ifakara, and Wanging'ombe districts, Tanzania. They established the peak adoption levels ranging from 29% (Ifakara) to 38% (Wanging'ombe) in 7–9 years, respectively. [19] used ADOPT to simulate the peak adoption level of post-rice legume crop in Southeast Asia. The results revealed the peak adoption of 54% in 6 years. Thus, this study uses the ADOPT to predict the adoption of Gliricidia agroforestry and fertilizer technologies in the study sites.

### 2.4. Conceptual Framework

ADOPT employs adoption theories and literature to provide an operationalized conceptual framework of factors influencing the adoption behavior of farmers in project communities [20]. The adoption of agricultural innovations is explained by economic drivers including profitability, risk-related factors, social context, and extent of farmers' engagement in testing technology under consideration [5,16,47]. In the ADOPT tool version for smallholder farmers, factors influencing the adoption of technologies in farming communities are categorized into two main groups: the relative advantage of technology and the effectiveness of the process of learning about technology [21,48].

Further, the relative advantage and learning factors are categorized into specific technology and population characteristics of the targeted farming community. At some points, population and technology variables may have linkages. For instance, the relative advantage of Gliricidia agroforestry technology may depend on its environmental benefits (a characteristic of the technology). However, the value of the environmental benefits of Gliricidia agroforestry technology depends on farmers' attitudes towards various environmental benefits (a characteristic of the population). The relative advantage and learning factors are combined with their corresponding technology and the population characteristics to form four sets of issues (here referred as quadrants) that are considered for increased adoption of improved agricultural technologies such as agroforestry (Figure 2). The two 'learning' quadrants on the left hand are (1) population-specific influences on the ability to learn about the technology and (2) the learnability characteristics of the technology. Population-specific learning factors at the top left quadrant are farmers' access to advisory support, group involvement, relevant existing knowledge and skills, and practice awareness. The technology-specific learning factors at the bottom left quadrant are observability, trialability, and innovation complexity (see Table 1 for descriptions). According to [49] the learning factors have no significant influence on the peak adoption level. However, the learning factors significantly influence the time to peak adoption [50]. This is because farmers take some time to learn about relevant information and experience before the subsequent adoption of the technology [20].

At the right are the relative advantages for the population and of the technology. Their factors are combined to determine the overall relative advantage of technology, which influences the peak level of adoption [49]. However, some aspects of relative advantage may also influence the time to peak adoption. For instance, when there is a high profit and environmental advantage, learning of the relative advantage of the technology becomes easier and more rapid [29]. Figure 2 presents the schematic conceptual framework of the organized ADOPT variables. Table 1 includes the range of question responses for each variable and the corresponding numerical codes. The latter serve as ADOPT inputs and feed into a series of equations to generate adoption predictions for the targeted population of farmers in the project village communities. Relative advantage factors such as risks, environmental benefits, and profit determine the proportion of the target population that is likely to adopt the introduced technology. On the other hand, the learning factors such as awareness, group involvement, and access to the extension services and advisory support influence the time lag before adoption decisions are made [20].

*2.5. Sampling and Data Collection*

This study was conducted in three villages in Kongwa and Chamwino districts where dryland agroforestry technologies were introduced under the framework of sustainable intensification (SI) and Climate-Smart Agriculture (CSA) as part of the research activities of the Africa RISING, Trans-SEC and Building Capacity for Resilient Food Security Projects in Tanzania (BCfRFS). The study villages were Mlali and Laikala in Kongwa, and Ilolo in Chamwino districts. These villages were purposively selected by the agroforestry project based on their diversity of food systems.

Smallholder cereal food producers were selected based on five sets of criteria. The first criterion was the years of experience in either trying or using the technology. Farmers with at least 2 years of experience were chosen from the agroforestry farmers' list obtained from the ICRAF office, in Dar es Salaam, Tanzania. The criterion of at least 2 years of experience was used because Gliricidia trees take a minimum of 2 years to produce sufficient pruning biomass for observable environmental and economic benefits [51]. Therefore, it is assumed that farmers with at least 2 years of experience in either trying or using Gliricidia agroforestry technology can explain various aspects of the technology such as its beneficial roles in mitigating climatic risks and increasing crop yields. The second criterion was the participation in either trying or using the mineral fertilizer technology on their farms. These farmers were selected from the list of agroforestry farmers containing description of farmers' socio-economic characteristics including age, gender, and the use of the mineral fertilizer technology. The third criterion

was the years of living in the village. Farmers with at least 10 years of living in the study villages were selected from the list of agroforestry farmers. Farmers who have lived in the village for at least 10 years are assumed to have learned about the behavior of people in the society, farming systems, and existing institutions and structures such as farmers' groups and cooperatives. Moreover, farmers with at least 10 years of living in the study villages are assumed to have accessed information on trends in production and markets of agricultural commodities. In the context of Tanzania communities, significant changes in governance and leadership that influence changes in institutions and structures occur after 10 years when the second phase of the general election takes place. The fourth and fifth criteria were sex and age of farmers based on literature recommendations and empirical evidence on the influence of gender and age in agroforestry technology adoption decisions from the previous studies [52,53]. Similarly, the study on adoption prediction by [5], considered the representation of all gender and age groups. Therefore, female and male, young people (18–35 years of age), and adults (above 35 years of age) as classified in Tanzania age categories participated in the current study. This composition ensured the collection of gender and age responsive opinions on ADOPT model variables (Figure 2).

Focus group discussions were conducted with groups of selected farmers, as recommended and used in related studies on adoption prediction [5,18–20]. The decision on the size of the group and number of FGDs was based on the guidelines provided for standard FGD [54–56]. One FGD session comprised of around 8–10 farmers. In total, 15 FGDs, five in each village were conducted. The FGDs were conducted in venues convenient for farmers. These included schools, churches, and village offices.

The discussions were guided by a checklist developed to capture information on 22 ADOPT variables, and were moderated by researchers who abided by the principles described in [56]. The Duration for one FGD interview was 3–4 h. The first 1.5–2 h, involved a detailed discussion, as strongly recommended for active participation of FGD members especially elder people [56]. The remaining hours were used for participatory coding of FGD responses based on the information provided by FGD participants. Similarly, [5] employed participatory approach to code FGD information for predicting the adoption of rainwater harvesting and fertilizer micro-dosing technologies during FGD.

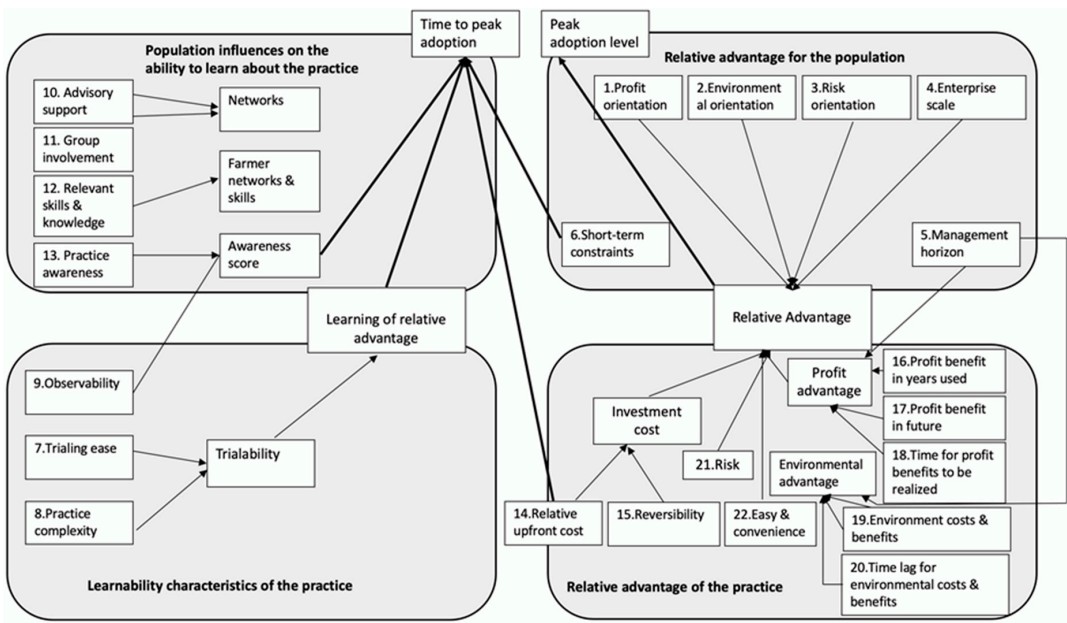

**Figure 2.** Adoption and Diffusion Outcome Prediction Tool (ADOPT) model variables. Source: Adapted from [20].

**Table 1.** ADOPT inputs.

| S/N | Variables | Question Asked | Measurement (Likert Scale: Statement of Minimum and Maximum Wording) | ADOPT Inputs/Mean Scores | |
|-----|-----------|----------------|----------------------------------------------------------------------|--------------|------------|
| | | | | **Agroforestry** | **Fertilizer** |
| 1 | Profit orientation | What proportion of the target population has maximizing profit as a strong motivation? | Almost none (1), almost all (5) have profit maximization motive | 4 | 4 |
| 2 | Environmental orientation | What proportion of the target population has protecting the natural environment as a strong motivation? | Almost none (1), almost all (5) have protection of environment motive | 3 | 3 |
| 3 | Risk orientation | What proportion of the target population has risk minimization as a strong motivation? | Almost none (1), almost all (5) have risk minimization motive | 2 | 2 |
| 4 | Enterprise scale | On what proportion of the target farms is there a major enterprise that could benefit from the practice? | Almost none (1), almost all (5) of the target farms have major enterprise that could benefit | 3 | 3 |
| 5 | Management horizon | What proportion of the target population has a long-term (greater than 10 years) management horizon for their farm? | Almost none (1), almost all (5) have long-term management horizon for their farms | 3 | 3 |
| 6 | Short term financial constraints | What proportion of the target population is under conditions of severe short-term financial constraints? | Almost all (1), almost none (5) have severe short-term financial constraints | 2 | 2 |
| 7 | Easiness in trialing | How easily can the practice (or significant components of it) be trialed on a limited basis before a decision is made to adopt it on a larger scale? | Not triable at all (1), very triable (5) | 3 | 3 |
| 8 | Innovation complexity | Does the complexity of the practice allow the effects of its use to be easily evaluated when it is used? | Very difficult (1), not at all difficult (5) to evaluate the effects of its use | 4 | 4 |
| 9 | Observability | To what extent would the practice be observable to farmers who are yet to adopt it when it is used in their district? | Not observable at all (1), very easily observable (5) | 4 | 4 |
| 10 | Advisory support | What proportion of the target population uses paid advisors capable of providing advice relevant to the practice? | Almost none (1), almost all (5) use a relevant advisor | 3 | 3 |
| 11 | Group involvement | What proportion of the target population participates in farmer-based groups that discuss farming? | Almost none (1), almost all (5) are involved in a group that discusses farming | 2 | 2 |
| 12 | Relevant existing skills and knowledge | What proportion of the community will need to develop substantial new skills and knowledge to use the practice? | Almost all (1), almost none (5) need new skills and knowledge | 3 | 3 |
| 13 | Awareness on the use/trialing of the innovation in the district | What proportion of the community would be aware of the use or trialing of the practice in their district? | It has never been used (1), almost all (5) are aware that it has been used/tried | 2 | 1 |

**Table 1.** *Cont.*

| S/N | Variables | Question Asked | Measurement (Likert Scale: Statement of Minimum and Maximum Wording) | ADOPT Inputs/Mean Scores | |
|-----|-----------|----------------|---------------------------------------------------------------------|:---:|:---:|
| | | | | Agroforestry | Fertilizer |
| 14 | Relative upfront cost of innovation | What is the size of the up-front cost of the investment relative to the potential annual benefit from using the practice? | Very large (1), no (5) initial investment required | 3 | 4 |
| 15 | Reversibility of the innovation | To what extent is the adoption of the practice able to be reversed? | Not reversible (1), very easily revised (5) | 3 | 4 |
| 16 | Profit benefits in the years that it is used | To what extent is the use of the practice likely to affect the profitability of the farm business in the years that it is used? | Large profit disadvantage (-3), very large profit advantage (+4) | 3 | 4 |
| 17 | Profit benefit in future | To what extent is the use of the practice likely to have additional effects on the future profitability of the farm business? | Large profit disadvantage (-3), very large profit advantage (+4) | 4 | 3 |
| 18 | Time for profit/benefits to be realized | How long after the practice is first adopted would it take for effects on future profitability to be realized? | More than 10 years (16), immediately (1) | 2 | 1 |
| 19 | Environmental impact | To what extent would the use of the practice have net environmental benefits or costs? | Large environmental disadvantage (-3), very large environmental advantage (+4) | 4 | 2 |
| 20 | Time for environmental impact to be realized | How long after the practice is first adopted would it take for the expected environmental benefits or costs to be realized? | More than 10 years (16), immediately (1) | 2 | 1 |
| 21 | Risks exposure | To what extent would the use of the practice affect the net exposure of the farm business to risk? | Large increase (-3), very large reduction in risk | 3 | 1 |
| 22 | Ease and convenience | To what extent would the use of the practice affect the ease and convenience of the management of the farm in the years that it is used? | Large decrease (-3), a very large increase in easy and convenience (+4) | 2 | 3 |

Data Coding and Entry

Focus group discussion information on each ADOPT variable was coded using either five or eight-point Likert scales. The corresponding mean scores were computed to generate ADOPT inputs for *Gliricidia* agroforestry and fertilizer use technologies (Table 1). Similarly, previous studies on adoption prediction used FGDs coded information to generate average scores which were used as inputs for ADOPT [5,18–20].

*2.6. Data Analysis*

First, the adoption and diffusion outcome prediction tool was used to predict peak adoption levels and time to peak adoption [20]. Data on the mean scores of ADOPT variables (Table 1), were fed into a series of equations (Table 2), with specific parameters' weights (Table 3) to predict the peak adoption level and time. The relative advantage variables such as the profit benefits in years used and profit benefit in future, were computed to predict the peak adoption levels. On the other hand, the learning variables like trialability and networks were computed to predict the time to peak adoption.

**Table 2.** Equations in the ADOPT model.

| **Peak Adoption** |
|---|
| Profit advantage = (Profit benefit in years used + Profit benefit in future × (1 + Discount rate)$^{-\text{Years to Future Profit Benefit}}$)/2 |
| Environmental benefit = $w_{eb}$ × Environmental benefit × (1+Discount rate)$^{-\text{Years to environmental benefit}}$ |
| Discount rate = 0.02 if Almost all have a long-term management horizon; 0.04 if A majority have a long term management horizon; 0.06 if About half have a long term management horizon; 0.08 if A Minority have a long-term management horizon; 0.1 if Almost none have a long-term management horizon. |
| Relative advantage = [(1 + $w_p$ × Profit orientation) × Profit advantage + (1 + $w_r$ × Risk orientation) × Risk + Ease and convenience + (1 + $w_e$ × Environmental orientation) × Environmental advantage] × (1 + $w_{es}$ × Enterprise scale) + $w_{ic}$ × (Investment cost − Max investment cost) |
| Peak adoption = $P_{min}$ +($P_{max}$- $P_{min}$)/(1+EXP($c_c$ − Relative advantage × $c_p$)) |
| *Time to Peak Adoption* |
| Trialability of Practice = (Trialing ease + Practice complexity)/2 |
| Networks = Min ($w_{gi}$ × Group involvement + Advisory support, 7) |
| Learning of Relative Advantage = Trialability of practice + Farmer networks skills + $w_{RA}$ × Relative advantage |
| Awareness Score = $A_{min}$+ Practice awareness + Observability − $A_o$ × Practice awareness × Observability |
| Farmer networks and skills = $F_a$ + $F_b$ × Relevant existing skills and knowledge + $F_c$ × Networks + $F_d$ × Relevant existing skills and knowledge × Networks |
| *Time to peak adoption* = MAX($T_{max}$ − Learning of Relative Advantage × $L_m$ + IF(UpfrontCosts ≥ 4, 0, $T_{min}$ − UpfrontCosts) + ($C_{max}$ − ShortTermConstraints) × ShortTermConstraints − AwarenessScore, 3) |

**Table 3.** Parameters' weights in ADOPT.

| | | | |
|---|---|---|---|
| $w_p$ | Profit Orientation Weight (0.4) | $C_{max}$ | Maximum Time Added Due to Short-Term Constraints (4) |
| $w_r$ | Risk orientation weight (0.2) | $w_{ia}$ | Practice awareness weight ("0") |
| $w_e$ | Environmental weight (0.4) | $w_o$ | Observability weight ("0") |
| $w_{ic}$ | Investment cost weight (0.33) | $A_{min}$ | Minimum level for awareness score (−1.25) |
| $w_{es}$ | Enterprise scale weight (0.4) | $A_o$ | Weight on interaction between practice awareness and observability (0.15) |
| $w_{re}$ | Risk effect weight (0.6) | $w_{eb}$ | Environmental benefits weight (0.6) |
| $T_{max}$ | Maximum time to adoption (50) | $w_{RA}$ | Rescales RA score to have equal influence on learning as do Trialability and Farmer Networks and Skills |
| $T_{min}$ | Minimum time to adoption (3) | $w_{gi}$* | Group involvement weight (0.7) |
| $P_{min}$ | Minimum adoption rate (1) | $c_c$ | Peak adoption curve parameter (3) |
| $P_{max}$ | Maximum adoption rate (98) | $c_p$ | Peak adoption curve parameter (0.3) |
| $F_a$ | Intercept term for Farmer networks and skills (-0.63) | $F_b$ | Weight on existing skills and knowledge (1.13) |
| $F_c$ | Weight on networks (0.63) | $F_d$ | Weight on interaction between networks and skills (−0.13) |
| $L_m$ | Scalar of Learning of Relative Advantage Score (3.0) | | |

Secondly, ADOPT conducted the sensitivity analysis to assess the effect of changes in scores of variables on step changes in adoption indicators: step-up and/or step-down responses of peak adoption level (percentage) and time to peak adoption (years). For the peak adoption level, the step up and step down responses are the respective increase and decrease in proportion of potential adopters of the technology. The vice versa is true for time to peak adoption: the increase in the predicted time to peak adoption presents the step down response, other factors remaining the same. Likewise, previous

studies on adoption prediction [5,18–20], conducted the sensitivity analyses to assess the effect of changes in scores of perception of ADOPT variables on step changes in peak adoption levels and the corresponding time, other variables remaining the same.

In this paper, results are presented using the Sigmoid (S)-curve based on the diffusion of innovations theory recommendations [21]. Moreover, in consonance with the diffusion of innovation theory, [57–59], argue that the S curve provides a good approximation of characteristics of cumulative adoption of the improved technologies. Similarly, the earlier studies on adoption predictions, [5,18–20], used the S curves to present the simulated ADOPT outcomes. In this study, the S curves were generated based on predicted values of peak adoption level and time to peak adoption, and starts from 2015, when the on-farm trials began.

## 3. Results and Discussion

### 3.1. Adoption and Diffusion Model Results

Results show that Gliricidia agroforestry technology has the peak adoption of 67.6%. On the other hand, the fertilizer use technology has the peak adoption of 34%. Moreover, results show more rapid adoption of Gliricidia agroforestry than fertilizer technology (Figure 3). Partly, the predicted higher peak adoption of Gliricidia agroforestry technology can be explained by farmers' attraction to the perceived relatively larger environmental benefits, future profit advantage, and the important role of Gliricidia trees in farm household strategies to reduce risk in agriculture (Table 1). Focus group discussion participants mentioned three major environmental benefits of Gliricidia agroforestry technology to their community. In descending order of importance, the environmental benefits reported by FGD participants are soil fertility improvement, rainfall facilitation, and soil erosion control. The adoption of Gliricidia agroforestry technology might be accelerated by the ongoing Government tree-planting campaigns in various communities including the study villages.

The relatively lower predicted peak adoption of the mineral fertilizer use in cereal production can be explained by the perceived higher risk exposure, smaller environmental benefits, and profit advantage (Table 1). Farmers use less or no mineral fertilizer due to the high probability of encountering huge losses in case of crop failure during drought. This agrees with Ellis's [60] concept that farmers applying large quantity of fertilizers are at risk of experiencing substantial profit loss during dry seasons, also referred as bad years. On the other hand, FGD participants reported that farms under agroforestry technology are not likely to be severely affected by drought. For instance, farmers reported to save up to 75% of crop yields from the effects of drought which occurred in the study sites in 2017. This can be associated with agroforestry systems ability to act as buffer against increased climatic variability [61–63].

Risk and economic factors have been pointed out by previous studies [5,9,14–16,39] as among factors contributing to low adoption of most agricultural innovations such as fertilizer use and agroforestry in SSA countries including Tanzania. However, [39,64], noted low use of fertilizer due to a lack of knowledge on use.

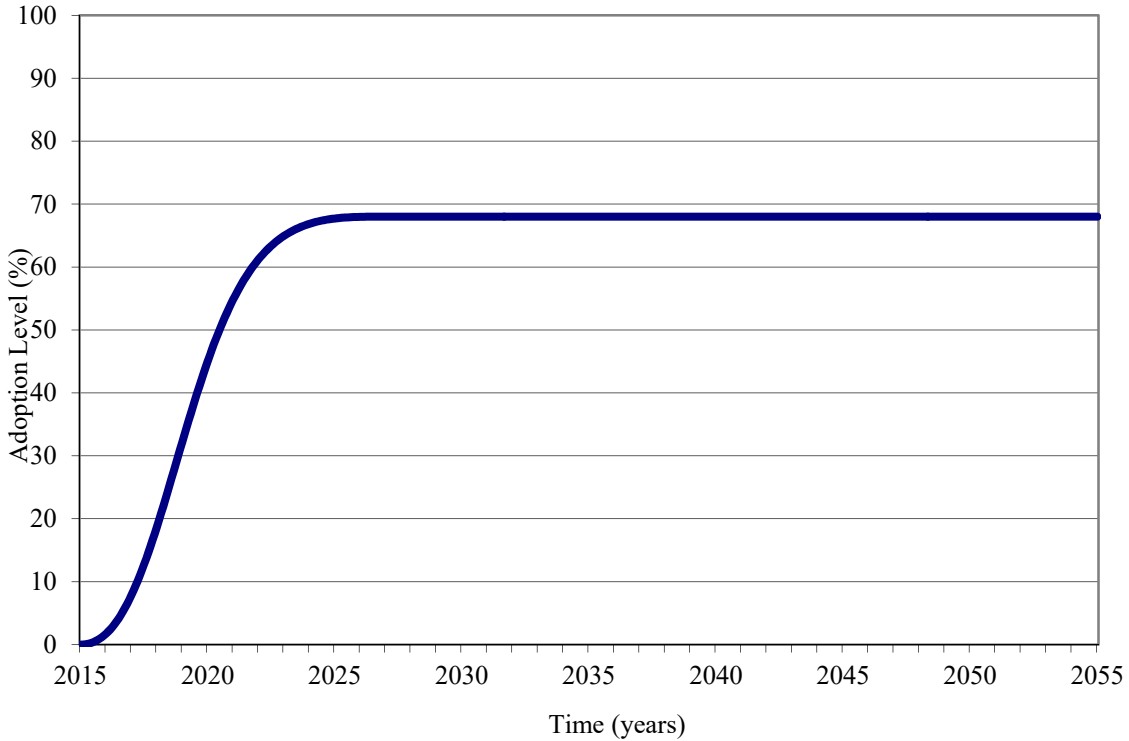

(**A**): Adoption of Gliricidia agroforestry technology.

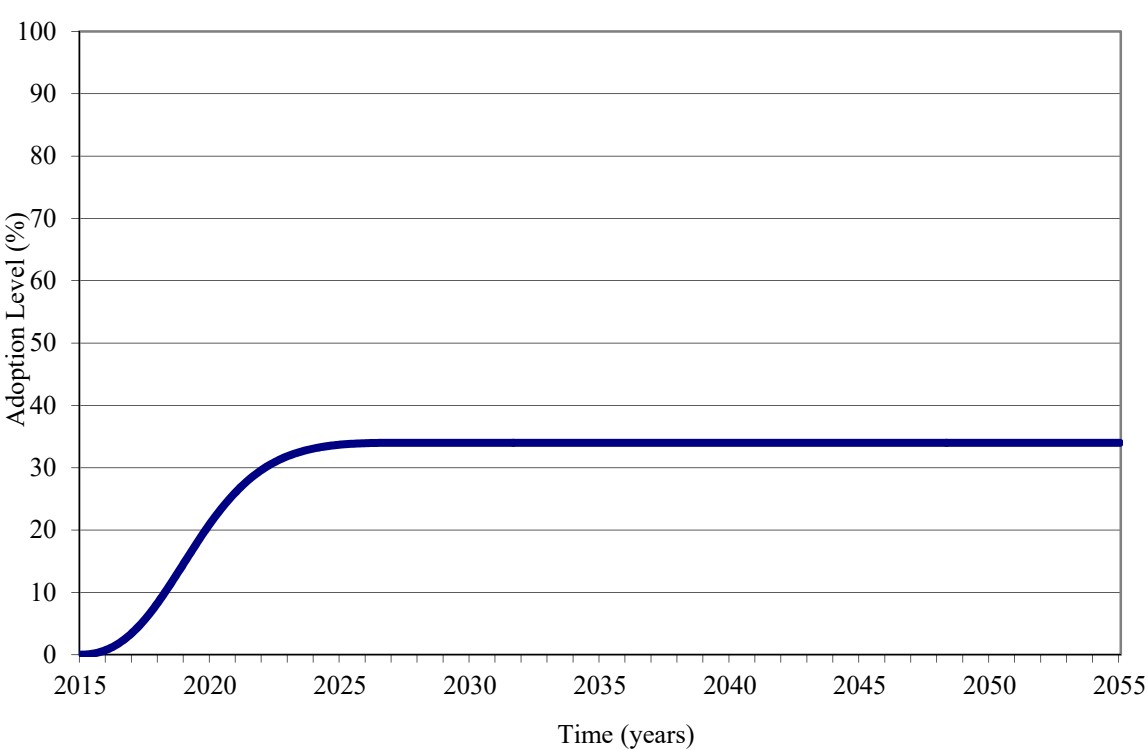

(**B**): Adoption of fertilizer use technology.

**Figure 3.** Adoption level Sigmoid (S) curves.

### 3.2. Sensitivity Analysis Results

Figure 4 presents the results of the sensitivity analysis of the changes in scores of perception of ADOPT model variables listed in Table 1. It shows that the adoption peak of Gliricidia agroforestry technology is highly sensitive to the upfront cost of agroforestry project establishment (14), environmental impact (19), risk exposure (21), and profit (16) (Figure 4). A unit decrease in score of perception towards a very large initial investment cost can step down the adoption peak level by 16.7%.

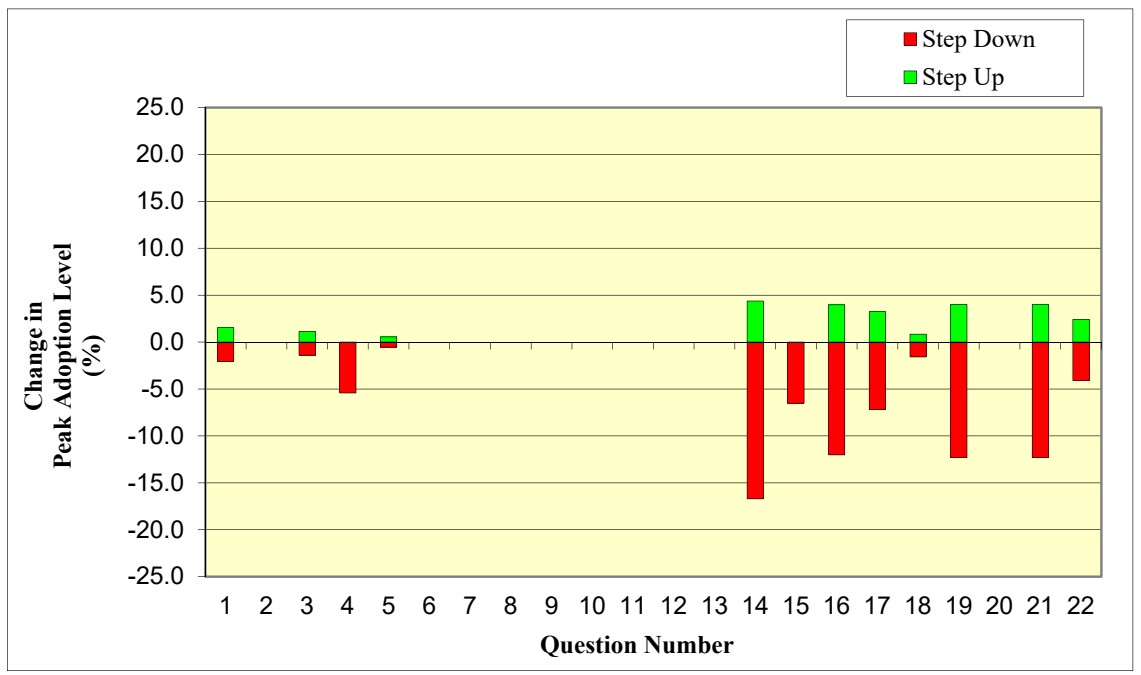

**Figure 4.** Sensitivity analysis to step-change in peak adoption level.

Previous studies by [5,9,65], observed the effect of the upfront cost on farmers' decision to adopt improved agricultural technologies such as agroforestry. According to these studies, the relative higher upfront cost for the establishment of the agroforestry project was attributed by the additional cost of planting materials and labor requirements in establishing tree nursery, transplanting, pruning of agroforestry trees, and application of pruning biomass into the soil. Also, [5,65,66] established the influence of capital requirement and profitability on the adoption rate of new agricultural improved technologies.

Further, results show that time to peak adoption is sensitive to major five factors. In descending order of magnitude, these factors are easiness in trying the technology (7), the complexity of the technology (8), short term financial constraints (6), existing skills and knowledge about the practice (12), and relative upfront costs of the practice (14; Figure 5). The results show that a unit increase in score of farmers' perceptions of easiness in trying the technology towards the very easy trialability, can reduce the predicted time to peak adoption level by 1.5 years and vice versa. Likewise, earlier adoption prediction studies by [5,18] observed the sensitiveness of trialability, short term financial constraints, and farmers' knowledge and skills on the time to peak adoption. According to [67], short-term loans play important roles in improving the livelihood of farmers, and in this case, could be the start-up capital for the establishment of the agroforestry project.

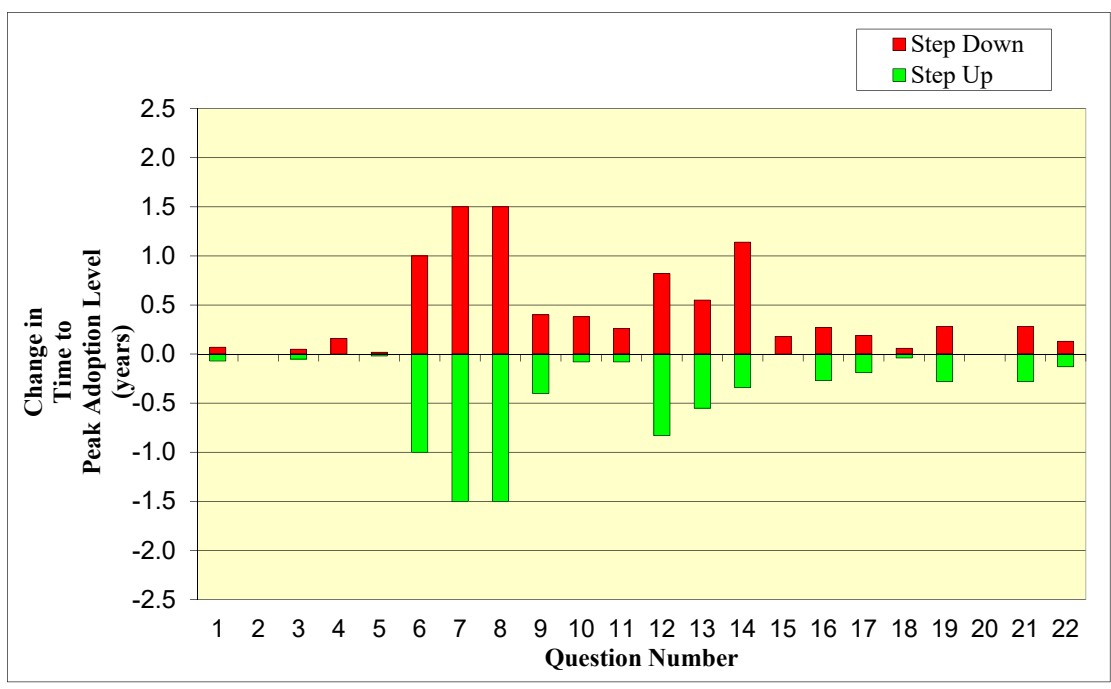

**Figure 5.** Sensitivity analysis to step-change in time to peak adoption level.

### 3.3. Conclusions and Recommendations

This paper predicted the adoption potential of Gliricidia agroforestry and the mineral fertilizer use technologies to inform policy and extension efforts on enhancing adoption of these technologies which works on complementarity.

The ADOPT results reveal that Gliricidia agroforestry has a relatively higher likelihood of being adopted than the mineral fertilizer use technology under current farming conditions. The higher adoption potential of Gliricidia agroforestry technology can be attributed to the perceived larger environmental benefits and profit advantage due to diverse products in addition such as crop yields and woodfuel. The environmental advantage of Gliricidia agroforestry based technology includes its roles in mitigating climate risks in agriculture, such as soil erosion control.

Moreover, this paper investigated the sensitivity of the ADOPT tool variables on the predicted adoption indicators of the peak level and time to peak adoption. The top five influential variables to the peak adoption level are upfront cost, reversibility, profit, environmental benefits, and risk.

Based on the results, it is concluded that Gliricidia agroforestry technology has high adoption potential and its adoption is influenced by the investment costs. We recommend two steps to enhance the adoption of Gliricidia agroforestry based technology. First, enhancing farmers' access to inputs, including tree seedlings, at affordable prices to lower the initial costs of investing in the agroforestry project. Second, increasing farmers' awareness of non-cash benefits such as soil erosion control besides the financial profit accrued after 1 year of agroforest project establishment. Increased awareness of non-cash benefits will attract cash-oriented smallholder farmers to invest in agroforestry projects. Awareness can be raised through the use of appropriate extension services, such as conducting farmer field schools and mobilizing farmers into groups.

**Author Contributions:** Conceptualization, M.S., D.P., A.M.A. and A.A.K.; methodology, M.S., D.P. and A.A.K.; software, M.S.; validation, M.S., D.P., A.M.A., S.S. and A.A.K.; formal analysis, M.S.; investigation, M.S.; resources, A.A.K.; data curation, M.S. and D.P.; writing—original draft preparation, M.S.; writing—review and editing, M.S., D.P., A.M.A., S.S., M.B., A.A.K.; visualization, M.S. and M.B.; supervision, D.P., A.M.A., S.S. and A.A.K.; project administration, A.A.K.; funding acquisition, A.A.K. All authors have read and agreed to the published version of the manuscript.

**Funding:** This research was co-funded by the United States Agency for International Development's (USAID) Feed the Future Project – Research in Sustainable Intensification for the Next Generation (Africa RISING)

(Grant No. AID-BFS-G-11-00002), the United States Department of Agriculture (USDA) with grant number FA19TA-10960C012, DAAD In-Country/In-Region funding programme managed through ICRAF (Grant No. 57300491), and the German Federal Ministry of Education and Research (BMBF) and German Federal Ministry for Economic Cooperation and Development (BMZ) through the Trans-SEC project (Grant No 031A249A). The APC was funded by the USAID Feed the Future Africa RISING project.

**Acknowledgments:** This paper is based on data collected from farmers involved in the technology testing and validation under the Africa RISING, Trans-SEC and BCfRFS projects. We are grateful to all who supported data collection, entry and processing, and to the anonymous reviewers and editors for constructive comments.

**Conflicts of Interest:** The authors declare no conflict of interest. The funders had no role in the design of the study; in the collection, analyses, or interpretation of data; in the writing of the manuscript, or in the decision to publish the results. The views expressed in this paper are purely those of the authors and may not under any circumstances be regarded as an official position of USAID, USDA, DAAD, BMBF, or BMZ.

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
