# Peer review of "Gliricidia Agroforestry Technology Adoption Potential in Selected Dryland Areas of Dodoma Region, Tanzania"

_agriculture, doi:10.3390/agriculture10070306_

Round 1
Reviewer 1 Report
I. Detailed comments
- Abstract:
- 27 Key processes making agroforestry of interest for agriculture are slow. The most important, increase of the organic matter content of the soil, takes so much time that two years of experience (1st selection criterion) is no experience.
- 28 One year of experience in using chemical fertilizer is at least as useless. At the start, fertilizer use if often inefficient, caused by the high C/N, C/P, …. Values of the soil organic matter. It causes a strong competition between soil (micro-)organisms and crops! Think also about the effects of soil and climate on fertilizer use efficiency!
- Introduction
- 62-64 Climate change does not have everywhere negative effects such as droughts. For example, for most of Tanzania, models predict higher rainfall.
- 78 Only in 2015 on-farm agroforestry tests have started. When did the present research start? What type of agroforestry has been tested (alley cropping, parkland, ….)?
- References 35, 36 & 37 without any indication of source of publication.
- 113-114 Hypothesis unclear.
- Methodology
2.1 Theoretical framework
- 128-131 The simple fact that group 1, the group of early adopters, is with 2.4% almost equal to Roger's 2.5% is not an argument for suggesting that Roger's theory applies here. Also 132-137 is not convincing. The theoretical framework ( 2.1) is weak.
2.2 Nature of technology change
- 142-143 How is "input" defined? A new technology, agroforestry in this case, is changing the input. The "same input" does not exist. Think for example about tree management! See 153-155!
- 148 Shouldn't "maximize" be "increase" or "improve"? Also in view of Pfister et al., 2005 a better wording.
- 152-153 "Optimize fertilizer use" a better description than "safe the costs".
- 156-157 In general does the presence of trees in the field decrease the crop yield with 20-30%.
2.3 Empirical review
- 181-189 The article of Kuehne et al. 2017 is not of interest as far as the predicted adoption rate and time for peak adoption concerns. It is the similarity between the model predictions and the observed adoption! This is the only real argument for the choice of ADOPT and the rejection of the other approaches. But only if the others have not been validated, or had validations with worse results.
2.4 Conceptual framework
- 221-224 "Time to peak adoption" will also be influenced in case of agroforestry by the rather long time required to observe positive effects from tree planting and management! Opposite suggestion 252-254
- 232 "schematic of the schematic" …..?
2.5 Sampling and data collection
- When the FGDs took place; how many years after start of the on-farm trials in 2015? Have the curves of fig. 5 been validated?
- 252-254 Two years sufficient for observable environmental and economic effects??? See 321-322.
- 254 Makumba et al., 2006 is missing under references. No verification possible.
- 254-257 What about the influence of the variability of the weather, of pests and diseases and other production factors? See 321-322.
- 256 Too optimistic! Farmers judging the beneficial effects of the trees they have just planted on the "mitigation of climate change".
- 257-258 It takes at least 5 years when farmers have a lot of knowledge and experience before fertilizer use efficiency on poor soils is reaching reasonable levels.
- 260-266 10 yrs also sufficient to know the agro-ecology of the village environment?
- 277 FGD should have been explained in OK, has been done already in Abstract.
- 276-278 15 x 3-4 hrs discussions with 8-10 farmers enough for adequate and precise answers? A reference should have been useful.
2.6 Data analysis
- 284 Data use; exploitation of data should be the title.
- 302-303. Agroforestry benefits are not visible at the start! See comments above. See first question under 2.5!
- Results and discussion
3.1 Adoption and diffusion model results
- 307-312 Has as much attention be given to the fertilizer use as to agroforestry? See 327-328! Effects of good fertilizer use much higher, much more visible that agroforestry effects. This is the more so at the start. By the way, who paid the costs, both of fertilizer use and agroforestry? Did farmers pay in both cases real prices?
- 322 "dry seasons" something different than droughts!
- 322-324 What about the risk of droughts when growing trees in the field?
- 333 Could the curves of fig. 5 been validated, at least as far as the first years concern?
- Have you additional arguments for trusting your results? Why I have encountered so many cases where farmers started to neglect the agroforestry as soon as projects stopped and project staff left?
3.2 Sensitivity analysis results
- 335-358 See last question i.r.t. 307-312.
- Why ADOPT used both for agroforestry and fertilizer adoption (5A and 5B) and the sensitivity test only for agroforestry?
- Impossible judging if the results have any value, without real quantitative data about key costs and benefits, both of agroforestry and fertilizer! Data about the region as well as data about Tanzania and other African countries. See for example From fed by the world to food security. "Accelerating agricultural development in Africa" and the example on p. 64 as far as the difference in fertilizer economics between 3 African countries concern.
3.3 Conclusion and recommendations
- Do your conclusions concern the villages, the region, the country, ….???? See last comment above.
- 365-367 See last two points under 3.1.
- 368-372 Are the results obtained enough for such a conclusion? See my questions.
- 379-381 And what would be the effect by doing the same with fertilizer costs?
- Serious editing required.
II. Comments and suggestions for authors
It will have been a real job organizing all those focus group discussions and translating farmers information into figures for the model. It is therefore that I regret not being more positive about the article. In brief, too much information is missing to be able to judge the general interest of the produced results.
- It does not become clear which agroforestry system has been studied. I presume that it concerns alley cropping with pruning of Gliricidia. But no information is presented about the conditions encountered by the participating farmers during the years of experience. Different soils? Rainfall and other weather conditions? Input and crop prices, labor availability and costs, and the variation over the years, and so on.
- It does not become clear how the agroforestry has been promoted. Has it only be through learning at the research stations that the farmers became acquainted, or did scientists and/or extension workers visited them also on their own farms? Did they receive manuals in the local language?
- How many years after the start of the project in 2015, the present research has been executed? Why not mentioning the adoption rate at the end of your research as validation for the curves obtained.
- Even less than about the agroforestry promotion is presented about fertilizer use promotion. The paper gives the impression that you believe more in Gliricidia than in fertilizer and I am therefore not astonished that you obtain a curve with a lower peak adoption.
- Did farmers obtain experience with both technology at real upfront costs and annual input costs, in particular as far as trees, fertilizer and labor concern? Have the input prices been stable since the start of the tests? What about the crop prices?
- Do you speak in your conclusion about local policy makers and extension workers, or in the district or at (inter)national level? What about the variation of all variables, of which I mentioned only some? Is your result already of interest?
- I presume that your conditions have been rather particular, in view of the fact that fertilizer worldwide and even in Africa is much more general than agroforestry adoption. If you are convinced that this trend should be changed and you like convincing others, you have to give much more attention to input and output markets, to agricultural policies and agro-ecological conditions.
Author Response
Thank you for your constructive inputs and comments. Kindly find the responses in the attached document

Reviewer 2 Report
Agriculture824865
Title: Gliricidia agroforestry technology adoption potential in selected dryland areas of Dodoma region, Tanzania
Authors: Martha Swamila
General Comments
The paper describes a study of designed to help implement agroforestry in Tanzania. The overall idea is good, and parts of the paper are interesting.
However, one concern is that the actual field methods are not described in enough detail. The data come from focus groups, but this is described in two short paragraphs.
A key part of the study is a sensitivity analysis of the data. This is interesting but described in one sentence. The description needs much more detail.
Also, the data used to generate the findings, presented in Figure 5, are not given. A summary of the data is needed. I see the Appendix, but it needs to be summarized concisely.
Therefore, this is an interesting idea but details on methods and findings need to be in the paper.
Specific Comments
- The Abstract is a bit too long. It reads like a paragraph. Try to reduce the length by 40 to 50%. The abstract should convey the most important points: why the study was done in one to two sentences; how the study was done in one to two sentences; the major findings; and, the implications. A good abstract is less than 300 words.
- The Introduction reads well. However, the last sentence, the hypothesis, needs more work. It is not a true hypothesis, which must be testable and falsifiable. Rather it is merely an objective, i.e., that knowledge might increase adopting the technique. Also avoid one sentence paragraphs. A paragraph is an idea that needs justification, which require more than one sentence.
- Parts of the Methods section belong in the Introduction. For example, the paragraph between line 179 and 194 is justification for the methodology, and thus it belongs in the Introduction.
- The subsection on Conceptual Framework, 2.4., also belongs in the Introduction. Another option is to make it a separate section, i.e., Introduction, Conceptual Framework, Methods, etc.
- Overall the Methods section is too long in aspects related to justification yet too short in the description of the data collection. You need to describe the focus groups survey, lines 276 to 283, in much more detail. Provide enough information for a reader to repeat the same study. Focus groups can give very biased findings depending on how the group was convened and treated.
- Also, the sensitivity analysis seems interesting, but you need to describe the approach. Again, describe in enough detail for a reader to repeat the study.
- I suppose the S-curve is okay. However, there is a huge literature on technology transfer and models. Did you consider alternatives? cf., Geroski, P. A. (2000). Models of technology diffusion. Research Policy, 29(4-5), 603-625.
- Where are Figure 3 and 4? Something is missing.
- It is not clear how you generated the s-curves in Figure 5. This needs to be explained.
Technical Comments
- Line 58: this is a run-on sentence. Divide into two sentences.
- Line 61: report percentages as integers. It is impossible to measure 0.1%
- Line 118: delete the reference to Yengoh et al. and USAID and reorganize the sentence. For example, say 'Rogers (2003) and Bass (1969) provide useful approaches, etc.'
- Line 119: delete the sentence 'these theories, etc.'
- Line 195: the sentence needs more detail.
- Line 287 and elsewhere: spelling, 'Likert'.
Author Response
Thank you for your constructive comments on the manuscript. We hope that we adequately responded to all of your helpful and insightful comments and critiques. In particular, we would like to draw your attention to the more detailed field methods i.e focus group discussion and the re-organization of the manuscript, including shifting of subsections i.e theoretical and conceptual frameworks from methodology to the introduction sections. All track change comments were edited directly in the manuscript. Our responses to specific reviewer comments are below. Note responses are the bolded words in italics.

Round 2
Reviewer 1 Report
Indeed, the authors made a serious effort to take my comments and suggestions into account. Reading the revised version and their reactions, I am realizing that differences in -sometimes subjective- judgements between the authors and myself cause irritations on my side. It makes me less objective in my judgement. However, the authors make in the new version and in their comments very clear that they are only trying to estimate the potential adoption. I hope that time will learn that they do it well.
Author Response
We would like to thank you for your insightful comments on our manuscript, as these comments led us to an improvement of the work
Reviewer 2 Report
I reviewed the original version. Thank you for addressing my comments and concerns.
Author Response
Thank you for taking time to review our manuscript. Our revisions reflect the constructive inputs and comments you made to improve our work